# Retinitis Pigmentosa (RP): The Role of Oxidative Stress in the Degenerative Process Progression

**DOI:** 10.3390/biomedicines10030582

**Published:** 2022-03-02

**Authors:** Enzo M. Vingolo, Lorenzo Casillo, Laura Contento, Francesca Toja, Antonio Florido

**Affiliations:** Department of Biotechnology and Medical-Surgical Sciences, “Sapienza” University of Rome, 04100 Latina, Italy; lorenzocasillo@yahoo.it (L.C.); laurapigreco@gmail.com (L.C.); francesca.toja@uniroma1.it (F.T.); antonioflorido@hotmail.it (A.F.)

**Keywords:** retinitis pigmentosa, oxidative stress, retina, circulation

## Abstract

Purpose: Retinitis Pigmentosa is a term that includes a group of inherited bilateral and progressive retinal degenerations, with the involvement of rod photoreceptors, which frequently leads to blindness; oxidative stress may be involved in the degeneration progression as proposed by several recent studies. The goal of this study is to evaluate whether circulating free radicals taken from capillary blood are related to one of the most important features of Retinitis pigmentosa that can affect frequently patients: cystoid macular oedema (CME). Materials: A total of 186 patients with Retinitis Pigmentosa (range: 25–69 years) were enrolled; all patients completed an ophthalmologic examination and SD-OCT at baseline and were divided into three subgroups according to the SD-OCT features. ROS blood levels were determined using FORT with monitoring of free oxygen radicals. Results: Test levels of free oxygen radicals were significantly increased, almost twice, in RP patients showing cystoid macular oedema and significantly increased compared to the control group. (*p* < 0.001). Discussion: Our findings suggest that oxidative stress may speed cone photoreceptors’ morphological damage (CMT); because long lasting oxidative stress in the RP may cause oxidative damage, with animal models of RP suggesting this is a micromolecular mechanism of photoreceptors’ (cone) death, it can be similar to cone damage in human RP eyes. The limitations of this paper are the relatively small sample, the horizontal design of the study, and the lack of data about the levels of ROS in the vitreous body.

## 1. Introduction

Retinitis Pigmentosa is a group of inherited bilateral and progressive retinal degenerations that involves rod photoreceptors and frequently leads to blindness. Although its molecular gene alterations have been well investigated by several groups since 1980, until now, it remains unknown why so numerous and different gene mutations may determine a very homogeneous clinical pattern. Several mechanisms are in action, causing photoreceptor apoptosis in human and animal models and in transgenic species. 

In this view, we have to consider that the equilibrium between oxidative reactions and antioxidant activities represents one of the principal factors inducing the evolution of genetic diseases in relationship to the lifestyle-related environment. In particular, photoreceptors are susceptible to oxidative stress as well described by several papers in the literature [1,2,3,4]; the main energy source of the rods is represented by the oxidative metabolism of fatty acids [5]. Even if the main causes of rod death are genetic mutations, and more than 100 RP gene mutations have been identified, we have to consider that in real life, very frequently, patients having the same genotype (identical DNA mutation) show different genotypes. 

Moreover, various recent studies highlighted that a high level of reactive oxygen species (ROS) are present in RPE cells, the membrane’s phospholipids and fatty acids are their molecular targets. These oxidized molecules may modify phototransduction pathways, intraretinal neurotransmission, and gene expression [6].

Better understanding the damage mechanisms and molecular pathways involved, it is fundamental to try a therapeutic approach that may lead to sparing the photoreceptors and rescuing sight. 

Recently, different authors have proposed the important role of oxidative stress. Within the retina, energetic processes, necessary for phototransduction and Vit. A regeneration, that are metabolized at high speed and widely diffused in the outer segment of photoreceptors, cause a large number of free radicals that usually are neutralized by biochemical processes in the outer retina, the Retinal Pigmented Epithelium (RPE), and in the choriocapillaris layer through several metabolic processes and pathways.

A recent paper on clusterin [4] demonstrated its activity as a cytoprotective protein in oxidative stress-induced cell death in a hypoxia-induced injury model, where the retinal pigment epithelial (RPE) cells, as in Retinitis pigmentosa, drive photoreceptors’ degeneration. 

Moreover, the use of next generation sequencing devices (NGS) recently allowed identification of noncoding RNAs and their targets host genes, that are involved in several biochemical pathways that may be related to a reduced or null response to oxidative stress; this causes a cascade of phenomena that affects visual functions, determining synaptic impairment of neurotransmission, damage of the interphotoreceptor matrix, and, finally, affecting the inner blood–retina barrier. 

From a new perspective, we can imagine a therapeutic approach that can interfere with these signals leading to delayed retinal cell death for apoptosis. Newly published data suggest a relevant role in Retinitis pigmentosa etiopathogenesis and the progression of noncoding RNAs in determining the visual damage that finally causes blindness. ARPE-19 cell line studies showed the possibility of finding a pharmacological solution to retinal dystrophies. The same cell line was investigated to identify a new candidate drug to protect RPE against oxidation [7]. 

Our goal in this study is to evaluate whether circulating free radicals taken from capillary blood may be related to one of the most important features of Retinitis pigmentosa that can affect frequently patients: cystoid macular oedema (CME). We have chosen this feature for following reasons:Usually macula (in which cones are prevalent) is less affected than peripheral retina, also in the late stages,It is easy to measure in vivo with an objective test (Optical Coherence Tomography).

## 2. Materials and Methods

In total, 186 Retinitis Pigmentosa patients (range: 25–69 years) were enrolled from the Rare Eye Disease Center at Terracina University Hospital. These patients were randomly selected and progressively numbered as they presented to our observation; we used the online randomization sequence generated by the APP (http://www.graphpad.com/quickcalcs/index.cfm, accessed on 6 December 2017) to obtain random numbers; then, we randomly assigned subjects to groups to undergo our observational study.

Each participant gave written consent before enrolment in the study. Institutional board approval was obtained from the ethics committee of our institution. All procedures were performed according to the suggestions of the Declaration of Helsinki.

Eligibility criteria included: retinal features of Retinitis Pigmentosa (bilateral extinct scotopic ERG, constricted Visual Field, and pigmented retina), visual acuity better than 20/200, a normal lifestyle based on a Mediterranean diet, mild to moderate physical exercise, abstinence from alcohol and tobacco products, no systemic diabetes, hypertension, or dyslipidaemia, and no excessive exposure to the light or ultraviolet rays.Exclusion criteria were: presence of coronary artery disease (CAD) or family history of CAD, prior treatment with Vitamin C or any other antioxidant dietary supplements, steroid or FANS drugs, history of acute or chronic eye infections, fever, cancer or organ failure, peripheral vascular disease, and thrombo-embolic events. We also excluded subjects with other ocular pathologies, severe dioptric media opacities, patients who had a history of intravitreal injection therapies and previous laser treatment, and patients within six months of intraocular or vitreoretinal surgical procedures.

All patients underwent a complete ophthalmologic examination and SD-OCT at baseline. Fluorescein angiography (FA) was performed in selected cases. Only one eye was considered for the study; if ME was present bilaterally, only the left eye was chosen.

A capillary sample was collected for free oxygen radical test (FORT) examination at the time of the visit before the uptake of the OCT scans. Spectral domain optical coherence tomography scans were performed with the Spectralis® HRA+OCT (software version 5.4.7.0, Heidelberg Engineering, Heidelberg, Germany), in a pattern of 20° × 15° (5.8 mm × 4.3 mm) raster scans consisting of 19 high-resolution line scans, each composed of 50 averaged frames. CMT was measured automatically using the standard protocols of the Heidelberg software. In addition, whenever both eyes of a patient were eligible, the right eye was selected as the study eye.

RP Patients were divided into three subgroups according to the SD-OCT features (Figure 1): Normal Retinal Thickness (NMT), Cystoid Macular Oedema (CME) without diffuse retinal thickening (DRT), and CME with DRT (wDRT). DRT is defined as increased retinal thickness (>200 μm) with reduced intraretinal reflectivity, especially in the outer retinal layers [8,9,10,11]. SD-OCT examination was performed by the same investigator. A group of 100 normal healthy patients was drafted as a control.

### 2.1. Oxidative Stress Analysis

The blood levels of ROS were determined using FORT with a free oxygen radicals monitor and kit (FORM®, CR 2000, Callegari, Italy). We used the FORM system to determine the concentration of ROS by a specific photometric kit of reagents called FORT. This is a colorimetric test based on the ability of the transition metals, such as Fe^2+^ or Fe^3+^, to catalyse the formation of free radicals in the presence of hydroperoxides (ROOH) (Fenton reaction). Radicals are then chemically trapped in an amine derivative.

Fenton reaction:Fe^2+^ + H_2_O_2_ → Fe^3+^ + OH + OH^−^.

This chemical reaction determines the formation of a more stable radical coloured cation, measured photometrically at 505 nm using the FORM analyser. Different intensities of colours are directly related to the quantity of radical compounds and the oxidative status of the blood sample. The ROS concentration is expressed using the free oxygen radical test (FORT) units (UF), one UF is approximately equal to 0.26 mg/L [7,12].

A volume of 20 μL of whole blood was drawn in the morning from each patient, after fasting overnight and before the use of any other medications. The blood samples were allowed to clot for 30 min; then, they were centrifuged at room temperature for 10 min. Finally, the samples were stored at −20 °C until the time of assay. For each measurement, three samples were used, and the mean value between all measures was assumed as the actual. The results obtained were expressed in FORT units [7]. The patients did not modify their therapy and dietary habits during the whole course of the study.

### 2.2. Statistical Analysis

Data are expressed as means ± standard deviation. The normality of distribution for each variable was analysed using the Shapiro–Wilk test. Repeated-measures analysis of variance (ANOVA) over the 3 time points (baseline, 3 months, and 6 months) and Bonferroni correction for multiple outcomes were used to determine whether any differences appeared in the best-corrected visual acuity (BCVA), expressed as log MAR units. 

The differences between groups were calculated using a 1-tailed unpaired *t*-test or Mann–Whitney U-test (nonnormal distribution, Shapiro–Wilk test *p* < 0.05). The Wilcoxon signed-rank and the Friedman tests were used to assess the changes over time when data were not normally distributed. Spearman’s rank correlation was calculated for comparisons. Intraclass correlations (ICCs) were computed to estimate the consistency when the same eye was evaluated multiple times with SD-OCT. The statistical significance was set at *p* < 0.05. All calculations were carried out with the use of SPSS statistical software (version 19; SPSS, Inc., Chicago, IL, USA).

## 3. Results

All data on oxidative stress value are summarized in the Figures; in Figure 2, they are expressed in conventional units (UF) and in Figure 3, in absolute value, expressed as mmol/L of oxygen peroxide (H_2_O_2_).

In the control group (100 patients), the oxidative stress index was 200.12 UF (±38.50 DS) and, expressed as concentration, was 1.53 mmol/L H_2_O_2_ (±0.29 DS).

In the Normal Retinal Thickness (NRT) Group (81 patients, 43.54%), the oxidative stress index was 249.37 UF (±45.65 DS) and, expressed as concentration, was 1.86 mmol/L H_2_O_2_ (±0.31 DS).

In the DRT Group (67 patients, 36.02%), the oxidative stress index was 355.79 UF (±139.26 DS) and, expressed as concentration, was 2.73 mmol/L H_2_O_2_ (±1.03 DS).

In the wDRT Group (38 Patients, 20.44%), the oxidative stress index was 339.47 UF (±115.20 DS) and, expressed as concentration, was 2.69 mmol/L H_2_O_2_ (±0.97 DS).

Free oxygen radical test levels were significantly increased almost twice in patients with retinitis pigmentosa showing cystoid macular oedema and significantly increased over the control group. *p* < 0.001). No statistical difference (*p* = 0.153) was found between the two groups with regional or diffuse macular involvement.

## 4. Discussion

Our findings suggest that oxidative stress may speed cone photoreceptor morphological damage (CMT) as previously suggested by Campochiaro [13,14]. As reported in the literature, ROS promotes damage of the blood–retinal barrier; modulating the production of vasoactive factors, ROS modifies retinal blood flow and also determines upregulation of the retinal expression of VEGF and other adhesion molecules such as ICAM-1 [15,16,17,18,19]. 

This cascade of molecular events causes leakage of fluids into the surrounding tissue, which initially accumulate within Müller cells. These fluids are clearly evident in OCT scans in the inner layers of the retina; moreover, the morphological changes precede the onset of the clinical features of CSME, are measurable by CMT, and may be defined as the DRT pattern. [3,4,8,9,15,17,18].

In experimental models of RP (mainly mice), the death of rod photoreceptors activates mechanisms that cause progressive oxidative damage directly to cones and, subsequently, follow the main channel of Muller’s architecture to other structural and glial cells of the inner retina [1,2,3,4,13,15]. 

Several studies in the literature pointed out the involvement of excessively high ROS concentrations and cellular damage in animals and in humans; in their eyes, high levels of proteins resulting from oxidative damage were detected in the aqueous humour, as compared to controls [4,16,19]. Moreover, cataract formation in the eyes may be related to oxidative stress and, in particular, to high levels of MDA in aqueous humour, as we demonstrated several years ago [20]. 

There is also evidence of chronic oxidative stress in comparison to healthy subjects, with a significant loss of GSH in aqueous humour recorded. This oxidative stress in patients with RP evaluated in blood cannot be justified by a contemporary systemic disease that also causes an increase in the oxidative level in the eyes, because serum proteins do not show an increase in oxidative damage; moreover, the serum GSH/GSSG ratio is unmodified, as is evident from different studies presented in the literature [21,22,23]. 

Oxidative stress and oxidative damage are also upregulated in RP retina, thus, suggesting that oxidative damage is a strong mechanism of cone cell death [1,3,8,13] in animal models of RP, and consequently may be applicable also to human photoreceptor loss in RP.

As shown in a previous study, total antioxidant capacity (TAC) was reduced in the aqueous humour of RP patients compared to controls [2,3,4,5,6,7,8,9,10,11,12,13,14]. Since GSH likely contributes to TAC, in our opinion, this reduced value of the GSH/GSSG ratio in RP patients is coherent with the observed loss in TAC. 

However, in the same study there were unclear data in the group of RP patients compared to controls; thio-barbituric acid reactive substances were significantly high in serum suggesting also that there may be oxidative damage to all lipids present in the entire body, which opens a new therapeutic dietary approach of the administration of a high amount of unsaturated fatty acid (DHA mainly) [1,14,16,18,20,21,22,23,24]. 

These data allow us to speculate that oxidative stress may be a strong biomarker for disease activity and macular involvement in each RP patient. In our opinion, to evidentially support this suggestion in patients with RP, it would be very important to turn the reactive oxygen species (ROS) measurement in peripheral blood into a long lasting clinical trial (at least three years), so that it would be possible, as a secondary endpoint, to test the effect of diet with high amounts of antioxidants on the functional outcomes RP [25]. 

In our previous study, we demonstrated that the use of steroids may improve the clinical conditions of cystoid macular oedema, as it is well known that this family of drugs counteracts biomolecular mechanisms involving proteins that are responsible for the low or incomplete neutralization of free radicals, as previously indicated (glutathione activity, scavengers, and others) [10,11]. 

Validation of these parameters as biomarkers for disease activity would greatly facilitate future clinical trials with a new dietary approach and partially explain the past studies of Berson with Vit. A and DHA and their beneficial effects on patients with RP [26]. Longitudinal studies in which serial measurements of ROS are correlated with the rate of reduction in the Goldmann visual fields and/or, with optical coherence tomography (OCT), reduction in the length of ellipsoid zone will also be interesting to evaluate.

The reduction in ROS levels leads to downregulation of the growth factors, adhesion molecules, and vasoactive substances and, consequently, reduction in vascular leakage. This is morphologically expressed by the CMT reduction that in our study was highly significant in patients who had taken antioxidant supplements.

These studies have several interesting “takeaways”: RP patients evidenced chronic oxidative stress and oxidative damage suggesting that demonstration of photoreceptor cell death due to oxidative damage in animal models of RP may also be applied to the human eye with RP [27,28,29,30].Data support moving forward with clinical trials to test whether dietary supplementation with antioxidants can slow cone photoreceptor loss of function and death in RP patients.Furthermore, our findings suggest that antioxidant supplements may improve morphological damage (CMT). Indeed, as reported in the literature, ROS promotes breakdown of the blood–retinal barrier and alters retinal blood flow by modulating the production of vasoactive factors, also upregulating the retinal expression of VEGF and adhesion molecules as ICAM-1 [31,32,33,34,35]. This cascade of molecular events causes leakage of fluids into the surrounding tissue, which initially accumulate within Müller cells. These morphological alterations may precede the appearance of CSME, they are quantifiable by OCT scans with CMT measurement, and they define the DRT pattern [4,8,11,15,36].Finally, measurement of ROS by the FORT technique is simple, fast, and affordable, as reported in previous studies, which assesses oxidative stress in patients with diabetes, cardiovascular diseases such as myocardial infarction, cancer, and other systemic diseases [20,21,24,25,37,38].The indirect measure of ROS was considered studying the effects of substances involved in the production or elimination of these species, using direct chemiluminescence and either direct or spin-trap electron paramagnetic resonance spectroscopy. Several reviews clearly described these methods, their needs and constraints, and their availability in the measurement of ROS [22,23,24,25,26,36,37,38,39].

## 5. Limitations

There were some limitations in this study as follows: The relatively small number of patients, considering that Retinitis pigmentosa is a rare disease, and this, also in consideration with the inclusion criteria, further reduces the possibility to include candidates;The horizontal design of the study that analyses a static environement. In our opinion, the test needs to be repeated over time, in a longitudinal study, correlating the value to the OCT situation. In consideration of the long disease duration (years), at least three years of observation would be necessary to evaluate the data more precisely.The present study, as others in the literature, draws on peripheral blood with consistent lack of data on the levels of ROS in the aqueous humour or in the vitreous fluid that are difficult to obtain in atraumatic way [15] in normal and RP eyes. These samples need to be acquired by anterior chamber paracentesis or pars plana vitrectomy; both are surgical procedures in which adequate consent could be difficult to obtain from the patients [40,41,42].

## Figures and Tables

**Figure 1 biomedicines-10-00582-f001:**
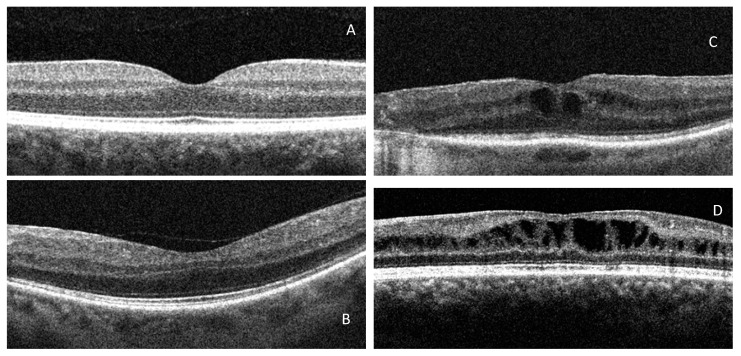
OCT groups: (**A**). Normal, (**B**). NMT, (**C**). DRT, (**D**). wDRT.

**Figure 2 biomedicines-10-00582-f002:**
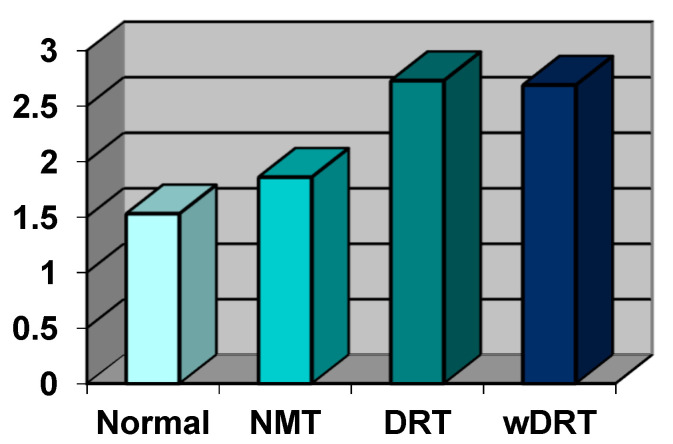
The value of the ROS species in all four groups of RP patients selected on the basis of OCT scans; data were expressed in free oxygen radical test (Fort) units. The value in the Normal Group was chosen as the reference.

**Figure 3 biomedicines-10-00582-f003:**
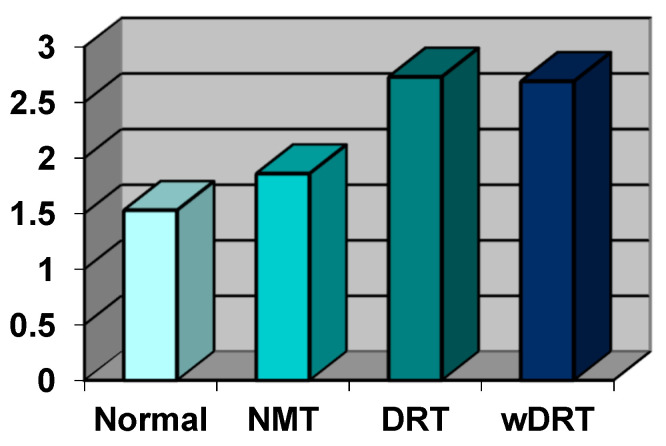
The value of the ROS species in all four groups of RP patients selected on the basis of OCT scans; data were expressed in mmol/L of the value of H_2_O_2_. The value in the Normal Group was chosen as the reference.

## Data Availability

Data were available in A.U.S.L Latina p.le P.L. Nervi 04100.

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
