# Peer review of "Retinitis Pigmentosa (RP): The Role of Oxidative Stress in the Degenerative Process Progression"

_biomedicines, 2022, doi:10.3390/biomedicines10030582_

Round 1

Reviewer 1 Report

Vingolo et al. produced a very interesting article describing the “Retinitis pigmentosa (RP) role of Oxidative stress”. I consider the manuscript very fascinating but, at the same time, I suggest several revisions needed to improve the reliability and the completeness of the paper:

  • The title should be more informative and revised in its English.
  • “Introduction” section: the “state of art” about oxidative stress and retinal diseases is not sufficiently updated. I suggest the authors to substitute and/or add more recent references related to innovative biomarkers of oxidative stress in retinal degenerations. For example, I suggest to change the references n° 3 and n° 6 with the more recent PMID: 32290199 and PMID: 32326576. Furthermore, the recent PMID: 34440511, PMID: 34058230 and PMID: 33801777 could represent a substrate able to enforce the role of oxidized stress within the onset or progression of inherited retinal diseases.
  • “Materials and Methods”: did the authors realize all experiments at least in triplicate?
  • Figures 2 and 3 are difficult to understand, due to a too short caption. Furthermore, is a standard deviation useful to be represented in shown bar plot or not? Thus, both figures should be improved.
  • “Limitations” should be put in another separate paragraph and better articulated.
  • Finally, manuscript requires English revisions and typos correction.

Author Response

  • The title should be more informative and revised in its English.
    • Title as changed including the degenerative process to better explicate the action of ROS studied
  • “Introduction” section: the “state of art” about oxidative stress and retinal diseases is not sufficiently updated. I suggest the authors to substitute and/or add more recent references related to innovative biomarkers of oxidative stress in retinal degenerations. For example, I suggest changing the references n° 3 and n° 6 with the more recent PMID: 32290199 and PMID: 32326576. Furthermore, the recent PMID: 34440511, PMID: 34058230 and PMID: 33801777 could represent a substrate able to enforce the role of oxidized stress within the onset or progression of inherited retinal diseases.
    • Introduction section was modified, and suggested references were added.
  • “Materials and Methods”: did the authors realize all experiments at least in triplicate?
    • For each sample measurement were performed on test, retest base and control, actual value was the mean between three tests.
  • Figures 2 and 3 are difficult to understand, due to a too short caption. Furthermore, is a standard deviation useful to be represented in shown bar plot or not? Thus, both figures should be improved.
    • Figures legenda has been modified, Standard Deviation value is well described in the text, presenting on bar graph may be, in our opinion a little confusing, adding more lines the readers cannot have clearly the idea of study results.
  • “Limitations” should be put in another separate paragraph and better articulated.
    • Limitation section was added
  • Finally, manuscript requires English revisions and typos correction.
    • Manuscript was reviewed by a separate from our group English native language person

Reviewer 2 Report

Dear authors:

Thanks for your working so hard to complete this excellent research.

The concept of study is very advanced and most of the readers wanted to know the results.

However, only one pity showed that your samples are from the blood other than the vitreous (at least draw the aqueous humor). Therefor, the higher oxidative stress parameters in the blood could explain the human body under oxidative stress, for example DM (from your reference 25).  The relationship between RP and oxidative stress is very weak. 

Could you supply any reference or your new description and let the readers clearly know?

Waiting for your positive response and thanks for give us so many.

Author Response

Thanks for your working so hard to complete this excellent research.

The concept of study is very advanced and most of the readers wanted to know the results.

  • However, only one pity showed that your samples are from the blood other than the vitreous (at least draw the aqueous humor). Therefore, the higher oxidative stress parameters in the blood could explain the human body under oxidative stress, for example DM (from your reference 25).  The relationship between RP and oxidative stress is very weak. 
    • As clearly stated in materials and Methods section, all patients with systemic diseases as hypertension, diabetes, dyslipidemia or coronary artery disease were apriori excluded from the study, also smoke and alcohol abuse did not entered in the research and finally dietary Mediterranean conditions were guaranteed for the inclusion in the study.
  • Could you supply any reference or your new description and let the readers clearly know?
    • New references were added.

Waiting for your positive response and thanks for give us so many.

Round 2

Reviewer 1 Report

The manuscript can be accepted in the present form.